# RELATIONAL REASONING AND INDUCTIVE BIAS IN TRANSFORMERS AND LARGE LANGUAGE MODELS

## ABSTRACT

Transformer-based models have demonstrated remarkable reasoning abilities, but the mechanisms underlying relational reasoning remain poorly understood. We investigate how transformers perform *transitive inference*, a classic relational reasoning task which requires inference indirectly related items (e.g., if $A > B$ and $B > C$, then $A > C$). Comparing in-weights learning (IWL) and in-context learning (ICL), we find that that IWL naturally induces a generalization bias towards transitive inference, despite being trained only on adjacent items, whereas ICL models trained solely on adjacent items do not generalize transitively. Mechanistic analysis shows that ICL models develop induction circuits that implement a simple match-and-copy strategy that performs well at relating adjacent pairs, but does not encoding hierarchical relationships among indirectly related items. However, when pre-trained on in-context linear regression tasks, transformers successfully exhibit in-context generalizable transitive inference, displaying both *symbolic distance* and *terminal item effects* characteristic of human and animal performance, without forming induction circuits. We extend these findings to large language models, demonstrating that prompting with linear geometric scaffolds improves transitive inference, while circular geometries impair performance, particularly when models cannot rely on stored knowledge. These results suggest that pre-training on tasks with underlying linear structure promotes the development of representations that can scaffold in-context relational reasoning.

## 1 INTRODUCTION

Transformer-based neural network architectures have been pivotal in recent advances in domains such as natural language processing (Vaswani et al., 2017) and computer vision (Chen et al., 2020), achieving human-like reasoning and generalization (Lake & Baroni, 2023). A key capability is their dual learning mechanism: storing information in weights during training (in-weights learning; IWL) and flexibly utilizing information from input sequences at inference time (in-context learning; ICL) (Brown et al., 2020). Given the prevalence and use of ICL, it is important to understand the inductive biases of ICL versus IWL (Lampinen et al., 2025; Dasgupta et al., 2022). Recent work using sequences of labeled examples $(x, f(x))$ has shown that ICL exhibits exemplar-based generalization while IWL shows rule-based biases (Chan et al., 2022a; Reddy, 2023; Singh et al., 2024).

While ICL has been studied extensively in classification and regression tasks (Reddy, 2023; Chan et al., 2022b; Singh et al., 2024; 2023), relational reasoning – integrating information across relationships to infer about indirectly related items – remains poorly understood. Recent work has begun investigating relational reasoning in large language models (LLMs), reporting varying degrees of performance (Liu et al., 2025; Li et al., 2024; Yang et al., 2024). However, for these LLMs it is difficult to study to what extent the model is using purely information from the context versus prior information stored in the weights due their complexity and to limited control of the training data. An open question, therefore, is how relational inductive biases differ between in-context learning and in-weights learning. To investigate this, we compared ICL and IWL in trained-from-scratch transformers on the fundamental relational reasoning task of *transitive inference* (Kay et al., 2024; Lippl et al., 2024) (Figure 1). Unlike category learning, which can rely on associative strategies, transitive inference requires understanding and applying ordered relationships in a hierarchy (e.g., if $A > B$ and $B > C$, then $A > C$). This allows us to probe the *relational* inductive biases in ICL and IWL.

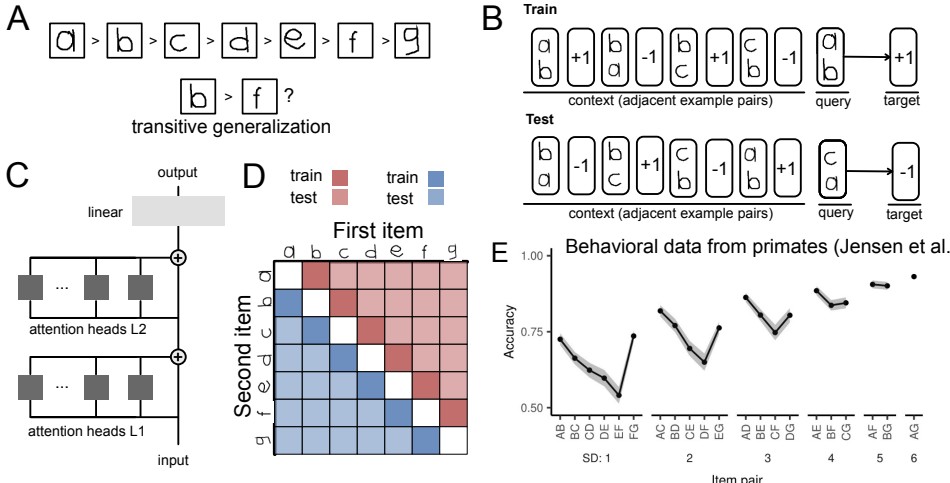

Figure 1: (A) Transitive inference setup with Omniglot images. First row shows an example hierarchy. Second row shows example evaluation of non-adjacent pair. (B) Transitive inference as a sequence. During training, the model is presented with a sequence defining the "hierarchy" (which items are larger than which), followed by a "query" consisting of an adjacent pair of items. The model is trained to categorize the order of the query pair: +1 if the first item is larger than the second, -1 if it is smaller. (C) Model architecture: two-layer attention-only transformer. (D) Illustration of the training set (adjacent pairs) and test set (non-adjacent pairs). Color indicates whether the first item is larger (+1) or smaller (-1). (E) Example accuracy on all training and test pairs (data from rhesus macaques). Figure reprinted from Lippl et al. (2024), data from Jensen et al. (2015).

Transitive inference (TI) tasks are widely studied in cognitive science and neuroscience (Vasconcelos, 2008; Nelli et al., 2023; Eichenbaum, 2004), where they reveal consistent behavioral patterns in TI behavior (Figure 1E). Firstly, performance generally increases with distance in the hierarchy (the "symbolic distance effect"; Moyer & Bayer, 1976). Secondly, some studies report better performance on training trials (symbolic distance of 1) than test trials with the same symbolic distance of 1 (the memorization effect; e.g. Nelli et al., 2023). Thirdly, performance is often better for trials involving *terminal* items, at either end of the hierarchy (e.g. Jensen et al., 2015).

Our main contribution is understanding how transitive inference (TI) capabilities differ across in-weights learning (IWL) and in-context learning (ICL) in transformers. We find that IWL naturally develops a transitive inductive bias despite training only on adjacent pairs, while ICL does not generalize transitively when trained on adjacent pairs alone. Mechanistic analysis reveals that ICL models implement match-and-copy operations through induction circuits rather than encoding hierarchical relationships – revealing a fundamental difference in inductive bias. However, pre-training on in-context linear regression tasks restores transitive inference in ICL without forming induction circuits, suggesting that exposure to linear structure during pre-training scaffolds relational reasoning. We validate these findings in LLMs, showing that linear geometric prompts improve transitive inference while circular prompts – which violate transitivity – impair performance, particularly when models cannot rely on stored knowledge. Both IWL and regression-pretrained ICL models recapitulate the symbolic distance and terminal item effects characteristic of human and animal performance. These results demonstrate that transformers' computational strategies for relational reasoning depend critically on their training regime and the structural biases they encounter during learning.

## 2 TASK AND NETWORK ARCHITECTURE

### 2.1 TRANSITIVE INFERENCE TASK STRUCTURE

Our TI task is based on a common in-context learning (ICL) setup where the model is presented with sequences of input-output pairs $(x, f(x))$. In our setup, each input token $x$ corresponds to a concatenated pair of adjacent items $x = \text{concat}(x_i, x_j)$, with the following label token $f(x)$ denoting

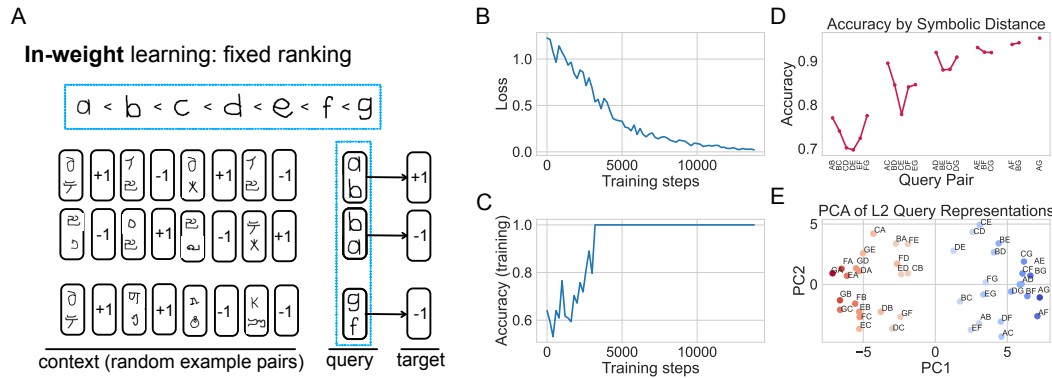

Figure 2: In-weights learning experiments. (A) Training and evaluation setup: the hierarchy is fixed across all sequences, and the context examples are randomly drawn and irrelevant for query prediction. (B) Training loss. (C) Training accuracy. (D) Final model accuracy for each pairwise comparison, sorted by symbolic distance. (E) Principal component analysis of the final hidden layer activations. Colors show signed symbolic distance from -6 (GA, red) to +6 (AG, blue).

if $x_i > x_j$ in the hierarchy (+1 if so, otherwise -1). Input tokens $x_i$ and $x_j$ are always adjacent pairs such that $|i - j| = 1$ Input features for $x_i$ and $x_j$ consist of pre-computed ResNet18 encoder embeddings of images from the Omniglot dataset (Lake et al., 2015), often used as a benchmark for few-shot or in-context learning (Chan et al., 2022b; Singh et al., 2024). The items and labels are embedded in $P + D$ dimensions, where the first $P$ dimensions encode positional information and the final $D$ dimensions encode the token's content. As in previous work (Reddy, 2023), position is encoded by a $P$-dimensional one-hot vector. $P = 64$ and $D = 1024$ for all experiments. We refer to the first $N$ item pairs and their labels as the *context*.

Following the context pairs, the model is given a query pair of items $\text{concat}(x_{q1}, x_{q2})$ and must predict whether $x_{q1} > x_{q2}$, by returning the correct label. The model is trained to minimize the loss on the query target prediction. Crucially, during training, only adjacent pairs of items (i.e., items that have a symbolic distance of 1) are used as queries. To evaluate the model's ability to perform transitive inference on indirectly related items, we present query pairs that are not adjacent, with a distance greater than 1 in the latent ranking.

### 2.2 IN-WEIGHTS VERSUS IN-CONTEXT PRE-TRAINING

**For IWL pre-training** (Figure 2A), we randomly draw $N = 7$ images from the dataset and assign each a fixed latent hierarchical rank $(A > B > C > D > E > F > G)$ that remains constant across training, enabling the model to learn these ranks in weights. The context contained $(N - 1) \times 2 \times 2$ tokens consisting of random unrelated image pairs with random labels, providing no information about the hierarchy while maintaining consistency with the ICL condition's context length. Each training sequence presented a query pair of two adjacent items from the fixed hierarchy, and the model was trained using MSE loss to predict the correct output (+1 if the first item ranks higher, -1 otherwise). Training proceeded for 14,000 iterations with batch size 128.

**For one-shot ICL pre-training** (Figure 3A), N items were randomly drawn from the dataset for each training sequence and assigned a latent rank. The context contains all adjacent image pairs in both orders $(N - 1) \times 2$, followed by tokens indicating their relative ordering. During training, the query was always one of these adjacent pairs. The latent rank was randomly assigned for each sequence, preventing the model from storing this information in weights and forcing it to infer the ranking from context to minimize training loss. Training proceeded for 40,000 iterations with batch size 128.

We also explored a different ICL approach, where we pre-trained the network on in-context linear regression problems. For this linear-regression pre-training, we followed (Oswald et al., 2023): the context tokens consisted of $N$ alternating $(x, f(x))$ pairs, where each $f(x) = Wx$. The true weights, $W$, are drawn randomly from a Gaussian for each sequence, meaning the model needs to use ICL

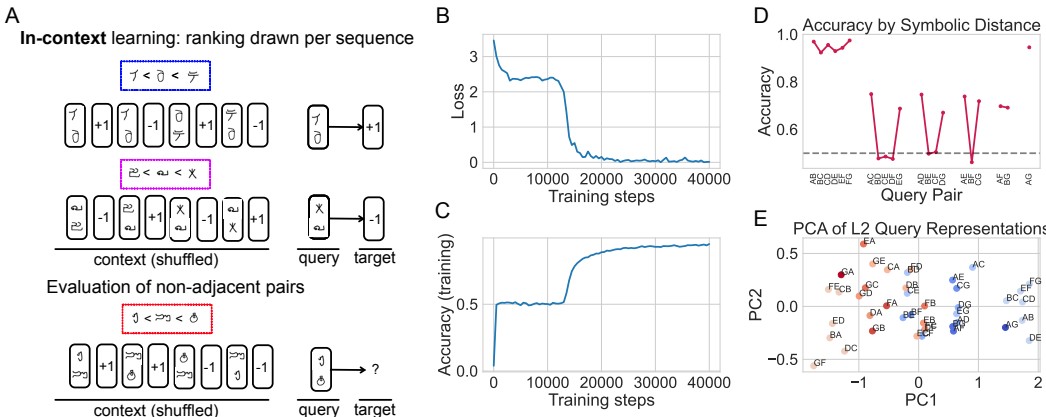

Figure 3: In-context learning experiments. (A) Training and evaluation setup. (B) Training loss. (C) Training accuracy. (D) Final model accuracy for each pairwise comparison, sorted by symbolic distance. Dashed gray line shows chance level. (E) Principal component analysis of the final hidden layer activations. Colors show signed symbolic distance from -6 (GA, red) to +6 (AG, blue).

to minimize training loss. Models underwent 50,000 training iterations with batch size 128. After pre-training, models were evaluated on the transitive inference task.

## 2.3 NETWORK ARCHITECTURE AND TRAINING DETAILS

For the main experiments, we used a two-layer attention-only transformer followed by a linear projection layer (Figure 1C). For each attention layer, each token $e_j$ is updated according to $e_j \leftarrow e_j + \sum_h P_h V_h \text{softmax} \left( K_h^T q_{h,j} \right)$, where $P_h$, $V_h$ and $K_h$ are projection, value and key matrices, $q_h$ is the query, and $h$ denotes the head index. The values, keys and queries are computed by linearly projecting the tokens. Here, each attention layer has 8 attention heads with a causal mask. In the standard transformer architecture, each self-attention layer is followed by a multi-layer perceptron (MLP), which we omit here for the purpose of interpretability, though we show experiments with the full transformer architecture in the Appendix.

## 3 RESULTS

### 3.1 TRANSITIVE INFERENCE EMERGES FOR IN-WEIGHTS LEARNING MODELS

Figure 2 shows the IWL results. At the end of training, the model exhibits transitive inference, performing above chance at all symbolic distances, despite having been trained only on adjacent pairs (Figure 2D). Furthermore, the model shows the "symbolic distance effect" previously shown in psychology experiments (Nelli et al., 2023): it performs better at longer symbolic distance. It also shows a strong "terminal item" effect, with better performance for trials involving terminal items (A and G), also often found in human subjects (Lippl et al., 2024). Interestingly, our IWL model did not show a clear "memorization effect" (stronger performance at distances of 1), despite being trained only at distances of 1.

We next investigated the representations that give rise to the TI behavior by performing principal component analysis (PCA) on the representations at the final layer of the final token in the sequence for all possible query pairs (Figure 2E). The first principal component (PC1) accounted for approximately 41% of the variance. We found that this PC1 could effectively separate positive and negative query pairs. Pairs were arranged continuously by signed symbolic distance, with adjacent pairs centrally located and distant pairs at the periphery, revealing the representational basis of the symbolic distance effect. The second PC ($\sim 11\%$ of variance) seems to reflect the identity of the second item. For low symbolic distances, item pairs ending on A, D or E score highly on this PC, item pairs ending with B, C or F score low, and item pairs ending on G sit around zero.

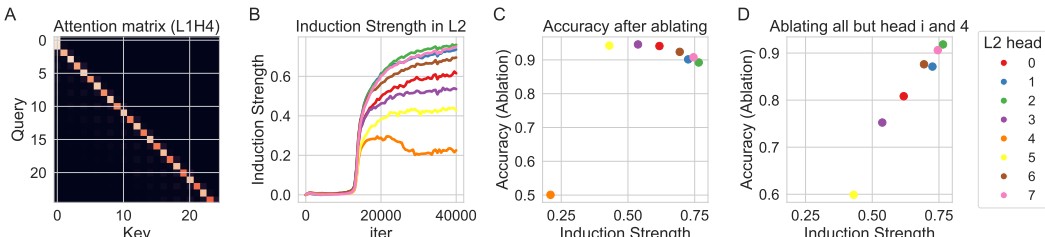

Figure 4: (A) Layer 1 attention pattern during the in-context TI task. (B) Induction strength of each layer 2 head during evaluation of the TI task with adjacent queries. (C) Induction strength versus accuracy after ablating head $i$. (D) Induction strength versus accuracy after ablating all but head 0 and head $i$.

### 3.2 Transitive inference does not emerge for in-context learning models

Figure 3 shows the results of learning exclusively in-context TI pre-training. In this ICL setup, we observed several key differences from the IWL results. First, we found a distinct phase change in the loss and accuracy curves, in accordance with previous studies on ICL (Figure 3B; 12000 steps). After the phase change, we found strong performance on sequences for which the query pair was adjacent and therefore present in the context (memorization effect). However, when inspecting the model's behavior on non-adjacent test trials, we found that the model exhibited no transitive inference (Figure 3D). We verified that this lack of transitive generalization was not due to the lack of MLPs in our attention-only model by also running experiments with the full transformer architecture (Figure A.4). In addition to the adjacent queries, the model performed above chance on queries containing one (e.g. AC or EG) or two (AG) terminal items. Note that this stronger performance on terminal items does not require transitive generalization, as it suffices to memorize that terminal items (i.e. items that only have one related item in context) are always on one end of the hierarchy.

This pattern reveals that ICL trained exclusively on adjacent pairs does not automatically develop transitive generalization, in contrast to IWL. Note that this reflects a difference in inductive bias rather than a fundamental limitation of ICL: when distal queries are added during training, our attention-only models are able to transitively generalize in-context when presented with sequences with novel stimuli, even though this does not show a symbolic distance effect (Figure A.5).

PCA analysis revealed no distance-like representations, unlike in the IWL case. Instead, while the first PC (explaining approximately 73% of the variance) separated the positive and negative train items, neither the first nor second PC (explaining approximately 4% of the variance) separated non-adjacent items which did not contain terminal items (for example, FD vs DF). These results suggest that the ICL model implements a fundamentally different computational strategy compared to the IWL model. Rather than learning relational structures that support transitive inference, the ICL model appears to rely primarily on matching of adjacent pairs, as well as memorizing terminal items A and G.

#### 3.2.1 Attention patterns reveal an induction circuit

Next, we asked what internal mechanism could give rise to the observed behavior of perfect training performance but lack of transitive inference and distance representations. Previous work has shown that transformers can learn to match and copy in-context through an "induction circuit" (Olsson et al., 2022), suggesting a potential explanation for our findings. Since our adjacent pair tasks can be solved through simple match and copy operations, we hypothesized that the network might be employing an induction circuit rather than learning the underlying transitive relationships. To investigate this, we analyzed attention patterns, revealing previous-token attention heads in layer 1 (L1) and induction heads in layer 2 (L2) (Figure 4A and B). For the layer 2 heads, we plot their "induction strength", defined as the difference in attention weights from the query token: attention to the correct in-context token - average attention to the incorrect in-context tokens. Figure 4B reveals that, concurrently with the sudden drop in in-context learning loss, each L2 attention head showed a sharp increase in induction strength, indicating the formation of an induction circuit.

To address whether the induction heads shown in Figure 4B were causally important for in-context prediction, we performed systematic ablations of each L2 attention head. We began by ablating each head individually, which revealed that head 4, despite having low induction strength, caused significant performance degradation when removed, while all other heads showed only minimal impact (Figure 4C). This finding is somewhat different from previous work in head pruning in different tasks, which has demonstrated substantial layer-wise redundancy where removing a single head typically produces minimal performance degradation (Michel et al., 2019; Voita et al., 2019; Singh et al., 2024).

To further investigate this pattern, we then performed systematic ablations where we removed all heads except head 4 and one additional head (head $i$). This analysis revealed a strong relationship between each remaining head's induction strength and its importance for prediction: performance scaled directly with the induction strength of head $i$. These results suggest a functional specialization where head 4 routes essential information (despite low induction strength), while the other induction-capable heads handle the pattern matching necessary for the adjacent pair task.

These findings indicate that the model solves the training task through specialized pattern matching mechanisms rather than by developing representations that encode the transitive relationships necessary for generalization.

### 3.3 Pre-training on linear regression yields emergent transitive inference during in-context learning

Next, given that our in-context learning task failed to produce emergent transitive inference, we investigated whether alternative pre-training approaches might yield different results. Previous work by Lippl et al. (2024) has demonstrated that transitive inductive biases can emerge from fundamental statistical learning principles, particularly norm minimization in linear models.

**Why does linear regression support transitive inference?** Linear regression induces an additive representational structure where each item is assigned an independent scalar "rank". When comparing items, the model learns that $A > B$ if rank(A) > rank(B). Norm minimization drives the model toward the simplest solution: a consistent linear ordering of all items rather than memorizing pairs independently. This means transitivity emerges automatically through the mathematical properties of real numbers: if rank(A) > rank(B) and rank(B) > rank(C), then rank(A) > rank(C).

We pre-trained our transformer model on an in-context linear regression task where input tokens consisted of alternating (X,Y) coordinate pairs (Figure 5A). This task was previously used by Oswald et al. (2023), who showed that transformers learn to implement gradient descent in the forward pass during ICL. This approach allowed us to evaluate whether exposure to explicitly linear relationships during pre-training would enhance the model's capacity to generalize transitively in subsequent tasks.

This pre-training strategy enabled the model to successfully exhibit transitive inference capabilities. The model also displayed both a symbolic distance effect, where accuracy increased with greater separation between items in the hierarchy, and a terminal item effect, with enhanced performance on items at the extremes of the sequence. These results suggest that pre-training on tasks with inherent linearity promotes the development of representations that capture transitive relationships, rather than the pattern matching strategies observed in our earlier experiments.

#### 3.3.1 Examining the attention patterns reveals no induction circuit

When analyzing the attention patterns of the linearly pre-trained model, we found no evidence of the induction circuit structure that characterised our previous model. Unlike the pattern-matching model, which exhibited clear previous-token heads in layer 1 and induction heads in layer 2, the linearly pre-trained model showed diffuse attention distributed across multiple positions and heads. These results suggest that the transitive inference capability emerged from distributed representations across the network rather than from specialized circuit components.

We also performed PCA analysis on the linear regression pre-trained model, finding that the first principal component explained approximately 72% of variance and separated item pairs by symbolic distance, similar to the IWL model. The second PC explains approximately 9.3% of variance. These results suggest that linear regression pre-training induces representations that capture hierarchical

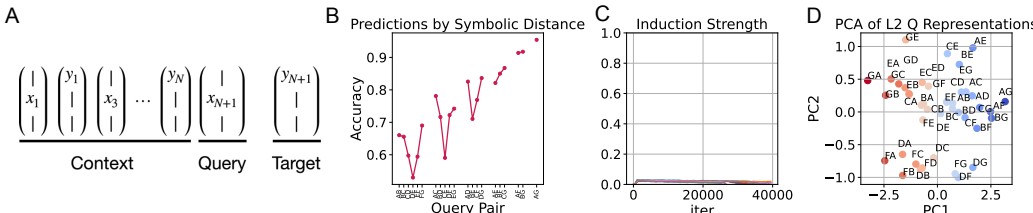

Figure 5: (A) Pre-training setup for in-context linear regression (evaluation setup was same as in Figure 3). (B) Post training model accuracy for each pairwise comparison, sorted by symbolic distance. (C) Induction strength of each head throughout training. (D) Principal component analysis of the final hidden layer activations. Colors show signed symbolic distance from -6 (GA, red) to +6 (AG, blue).

relationships, unlike the standard ICL model which relied on pattern matching without developing distance-based representations.

### 3.4 LINEAR GEOMETRY SCAFFOLDS IMPROVES TI IN LLMS

To investigate whether our findings extend to large-scale models, we conducted experiments with pre-trained large language models using the ReCogLab transitive inference dataset (Liu et al., 2025). This dataset generates reasoning problems with three types of item relationships: congruent with real-world knowledge (e.g., whale > dolphin > goldfish), incongruent with real-world knowledge (e.g., goldfish > dolphin > whale), and random strings with no semantic meaning. Liu et al. (2025) found superior performance on congruent items, which we hypothesized reflects the models' ability to leverage knowledge stored in weights rather than performing true in-context TI.

To probe the geometric inductive biases underlying transitive reasoning, we augmented the standard prompts (see Liu et al. (2025)) with two geometric scaffolds. The linear prompt instructed: "Imagine all of these items lie on a number line from smallest to largest," while the non-linear prompt used: "Imagine all of these items lie on a circle from smallest to largest." Critically, circular orderings violate transitivity: while $A > B$ and $B > C$ implies $A > C$ in a linear ordering, this does not hold on a circle where relationships can wrap around. Thus, we predicted that circular prompts would impair transitive inference performance. To isolate the effect of these geometric representations from chain-of-thought reasoning, we explicitly instructed models to answer with only "yes" or "no" and restricted output tokens (see Appendix).

Across all tested language models except Gemma 1B (the smallest model), we found that the linear number line prompt consistently outperformed the circular prompt (Figure 6a), confirming that the theoretically incompatible circular geometry indeed disrupts transitive reasoning. This disadvantage of circular geometry was most pronounced in the incongruent and permuted conditions (Figure 6c), where models cannot rely on pre-existing knowledge in weights and must perform genuine in-context transitive inference. In these conditions, the circular prompt's violation of transitivity maximally conflicts with the task demands. In contrast, for the congruent condition where stored knowledge aligns with the task, the difference between geometric prompts was minimal, and both prompt effects negative. In that condition, it is better for the model to ignore the geometric prompt and use stored knowledge. Figure A.2 shows these effects for each model separately. In Gemma models, the advantage of linear over nonlinear prompting increased with model size (Figure A.3).

These results parallel our findings from trained-from-scratch transformers: when models must rely on in-context learning rather than stored knowledge, the choice of representational geometry critically impacts transitive inference performance. The degraded performance with circular representations, which theoretically cannot support transitivity, demonstrates that transformers' success at transitive inference depends on adopting compatible geometric scaffolds. This provides important validation that our controlled findings generalize to modern large-scale language models

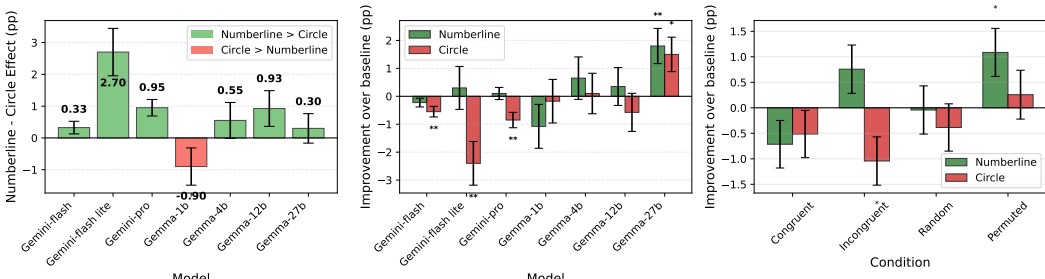

Figure 6: Comparative analysis of geometric prompting strategies on transitive inference tasks across language models. (a) Direct comparison of linear (number line) versus non-linear (circle) geometric prompts, showing the difference in accuracy improvement for each model. (b) Overall prompt strategy effects showing accuracy improvements over baseline for number line and circle prompts across all tested models. (c) Prompt effectiveness by item type (congruent, incongruent, random, and permuted conditions), showing larger improvements for number line prompts in incongruent and permuted conditions compared to congruent conditions. All improvements are measured in percentage points relative to baseline. Error bars represent SEM.

## 4 DISCUSSION

Our findings demonstrate a contrast between learning approaches in transformer models tasked with transitive inference. When trained to store adjacent pair relations in weights, the model successfully exhibited transitive inference. In contrast, models trained through in-context learning on adjacent pairs failed to develop this capability despite perfect training performance. Analysis revealed that these models relied on induction circuits for pattern matching rather than encoding hierarchical relationships. Pre-training on in-context linear regression tasks restored transitive inference abilities, producing both symbolic distance and terminal item effects.

### 4.1 INDUCTIVE BIASES ASSOCIATED WITH IWL AND ICL

Recent work has uncovered different inductive biases between learning algorithms. In category learning, some algorithms exhibit a 'rule bias' while others show an 'exemplar bias' (Dasgupta et al., 2022). Chan et al. (2022a) showed transformers operate in the rule-based regime when storing information in weights, but generalize by exemplar from context. Lampinen et al. (2025) found LLMs generalize more flexibly from in-context learning than from (in-weights) fine-tuning.

Transitive inductive biases have been studied in neural networks using IWL setups, but not ICL as done here (Battaglia et al., 2018). Nelli et al. (2023) showed in a neural network tailored for this task (as well as in human dorsal frontoparietal cortex) that representations for each item lie on a number line. Lippl et al. (2024) explained TI through ratios of conjunctive versus additive representations. Most similar to our work, Kay et al. (2024) trained recurrent networks on sequential TI tasks. Miconi & Kay (2025) trained a recurrent neural network on TI-sequences where a different ordering was drawn for each sequence, similar to our work. However, this setup differs from ours in two key ways: (1) the network performed weight updates within sequences and (2) the non-adjacent pairs were given at the end of each sequence. Swaminathan et al. (2023) showed that their clone-structured graph model exhibits transitive generalization during a sequence learning task, but did not explicitly test for symbolic distance or terminal item effects. Liu et al. (2025) investigated in-context TI in pre-trained large language models, but focused on model behavior without analyzing internal representations.

### 4.2 MECHANISMS OF ICL

Previous work has shown that the abrupt transitions in ICL correlate with the formation of induction circuits" Olsson et al. (2022); Singh et al. (2024); Reddy (2023): two-layer circuits consisting of a previous token head" which attends to the previous token, and an "induction head" in the next layer that performs match-and-copy operations. Previous studies have shown that abrupt changes in the loss function can be explained by different subcircuits of this induction circuit Reddy (2023); Singh

et al. (2024), similar to what we show here. Other work has hypothesized that transformers learn to implement gradient descent in the forward pass (Oswald et al., 2023) (but see Shen et al. (2024)). Our study provides empirical evidence illuminating the distinction between these mechanisms: when pre-trained on a copying task, the model developed clear induction heads, but when pre-trained on linear regression, these induction heads were absent. This suggests that the computational algorithm implemented by in-context learning is not fixed but adapts to the task demands. Recent work has similarly shown that ICL may involve multiple competing algorithms that shift based on data diversity and training dynamics (Park et al., 2025), consistent with our finding that different pre-training regimes yield qualitatively different computational strategies.

### 4.3 BRIDGING TRAINED-FROM-SCRATCH MODELS AND PRE-TRAINED LLMS

Our LLM experiments provide an important connection between our controlled studies and real-world language models. The advantage of linear over circular prompts suggests that LLMs have likely encountered ordering and regression-like tasks during pre-training, developing capabilities for linear relational reasoning. By prompting with linear geometry, we can activate these learned representations, while circular prompts, which violate transitivity, interfere with this capability. The detrimental effect of circular prompts validates our theoretical prediction: since circular orderings cannot support transitive relationships (where $A > B > C$ does not imply $A > C$ due to wraparound), models prompted with this incompatible geometry show degraded performance. The smallest model (Gemma 1B) uniquely showed a weak reversal. This likely represents a floor effect where models below a capacity threshold cannot effectively process these instructions. The fact that, within the open-weight models of known size, larger models show a bigger effect, supports this interpretation (Figure A.3). The minimal geometric effect in congruent conditions, where LLMs can leverage stored knowledge, parallels our IWL models that developed transitive capabilities through weight storage. However, in incongruent conditions requiring pure in-context reasoning, we observed a large difference between number line and circular prompts. These findings indicate that transformers' ability to perform transitive inference depends on accessing appropriately structured representations, whether developed through targeted pre-training (as in our regression experiments) or activated through geometric prompting in models that have encountered similar structures during training.

### 4.4 LIMITATIONS

First, while we extend our findings to large language models through prompting experiments, our mechanistic analyses (PCA, ablation studies, attention patterns) were restricted to small transformer architectures for interpretability. The internal mechanisms in LLMs performing transitive inference remain opaque, and future work should combine these experiments with advanced mechanistic interpretability techniques. Secondly, the transitive inference task we designed is a simple example of relational reasoning which might not capture all real-world scenarios, but it is well studied in cognitive psychology and neuroscience (e.g. Eichenbaum, 2004; Vasconcelos, 2008; Jensen et al., 2015; Nelli et al., 2023). The use of such cognitively-grounded tasks provides a principled approach to evaluating model capabilities (Rane et al., 2025), making it a well understood benchmark to study model behavior and representations. Thirdly, while our results are empirical rather than theoretical, we have strengthened our findings through systematic ablation studies and attention pattern analyses to provide mechanistic insights. These behavioral patterns and circuit-level observations can serve as grounding constraints for future work on understanding relational reasoning in transformers.

### 4.5 CONCLUSIONS

We have explored the transitive generalization abilities of transformers during in-weights learning (IWL) and in-context learning (ICL) with different pre-training setups. We find notable differences in both performance and underlying mechanisms. While IWL transformers naturally develop transitive inference capabilities, ICL only exhibits this ability when pre-trained on tasks with inherent linearity. Our mechanistic analysis reveals that these behavioral differences correspond to distinct computational strategies: pattern matching through induction circuits versus distributed representations that encode relational structure. Critically, we demonstrate that these insights from controlled experiments with small transformers generalize to modern large language models. These findings provide empirical insights into how different learning paradigms shape the inductive biases of transformer-based language models, with implications for understanding both artificial and biological learning systems.

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

# A APPENDIX

## A.1 DATASET AND PREPROCESSING

We used pre-computed ResNet18 embeddings of Omniglot images as input features. These embeddings were taken from a dataset provided by (Singh et al., 2023). All experiments employed hierarchies of $N = 7$ items (A, B, C, D, E, F, G). For each sequence, images were randomly sampled from the Omniglot dataset without replacement. Input tokens consisted of $P = 64$ positional dimensions (one-hot encoded) concatenated with $D = 1024$ content dimensions from ResNet18 embeddings, resulting in total embedding dimensionality of 1088.

## A.2 TRAINING PROCEDURES

All ICL models were trained for 40,000 iterations. IWL models were trained for 14,000 iterations. Context length was fixed at $(N - 1) \times 2 \times 2 = 24$ tokens for $N = 7$ hierarchies. We used mean squared error (MSE) loss between predicted and target labels.

For training, we use a batch size of 128 and the Adam optimizer (Kingma & Ba, 2015) with a learning rate of $10^{-3}$ and a weight decay parameter of $10^{-7}$. We selected these hyperparameters through manual tuning. All experiments were conducted on a single NVIDIA GeForce GTX TITAN X GPU with 12GB memory and did not last longer than 10 minutes per run. Additional training details are provided in the Appendix

### A.2.1 IN-WEIGHTS LEARNING (IWL)

The same 7 Omniglot images maintained fixed rank order across all training sequences. Context consisted of random unrelated image pairs that provided no information about the hierarchy, forcing the model to learn adjacent pair relationships through weight storage.

### A.2.2 IN-CONTEXT LEARNING (ICL)

Each sequence used new random rank assignments for the 7 images. Context contained all $(N - 1)$ adjacent pairs presented in both orders with correct labels. Queries consisted of one adjacent pair from the context to test memorization ability.

### A.2.3 LINEAR REGRESSION PRE-TRAINING

Models underwent 50,000 sequences of in-context linear regression pre-training. Context contained $N = 12$ $(x, y)$ coordinate pairs where $y = Wx + \epsilon$, with weight matrix $W$ randomly sampled per sequence from a Gaussian distribution and Gaussian noise $\epsilon \sim \mathcal{N}(0, 0.1^2)$ added to $y$ values. After pre-training, models were evaluated on transitive inference without additional training.

## A.3 MODEL ARCHITECTURE DETAILS

Our transformer employed 2 attention-only layers without MLPs or layer normalization. Each layer contained 8 attention heads with causal masking. The output consisted of a single linear projection to scalar output, followed by a sign function for classification accuracy. All parameters used PyTorch standard initialization.

## A.4 EVALUATION PROTOCOL

During training, only symbolic distance 1 (adjacent) pairs were presented as queries. Evaluation included all possible pairs with symbolic distances 1-6. We measured accuracy (fraction of correct higher / lower predictions) for each symbolic distance.

### A.5 ATTENTION ANALYSIS METHODOLOGY

For each attention head $h$ and query position, we calculated induction strength as [attention to the correct in-context token] - [average attention to incorrect in-context tokens]. Head ablation was performed by setting the target head's output to zero.

### A.6 PRINCIPAL COMPONENT ANALYSIS

PCA was applied to final layer representations of query tokens across all possible item pairs. Representations were centered but not scaled before analysis. Visualizations were colored by signed symbolic distance, distinguishing negative cases (first item < second item) from positive cases (first item > second item).

### A.7 HYPERPARAMETER SELECTION

Learning rate was manually tuned from $\{10^{-4}, 5 \times 10^{-4}, 10^{-3}, 5 \times 10^{-3}\}$ with $10^{-3}$ selected. Weight decay was tuned from $\{0, 10^{-8}, 10^{-7}, 10^{-6}\}$ with $10^{-7}$ selected. Batch size was fixed at 128 based on memory constraints. We used the Adam optimizer with PyTorch default parameters: $\beta_1 = 0.9$, $\beta_2 = 0.999$, $\epsilon = 10^{-8}$.

### A.8 COMPUTATIONAL RESOURCES AND REPRODUCIBILITY

All experiments were conducted on a single NVIDIA GeForce GTX TITAN X GPU with 12GB VRAM using PyTorch 1.12 and CUDA 11.3. Training time was approximately 5-10 minutes per experiment. Random seeds were fixed (seed=42) for reproducibility. While results shown are from single training runs due to our focus on mechanistic analysis, key findings (Transitive inference and symbolic distance effects for IWL and linear-pretrained ICL models) were replicated across 10 independent runs with different random seeds, demonstrating consistent large effect sizes (Figure A.1).

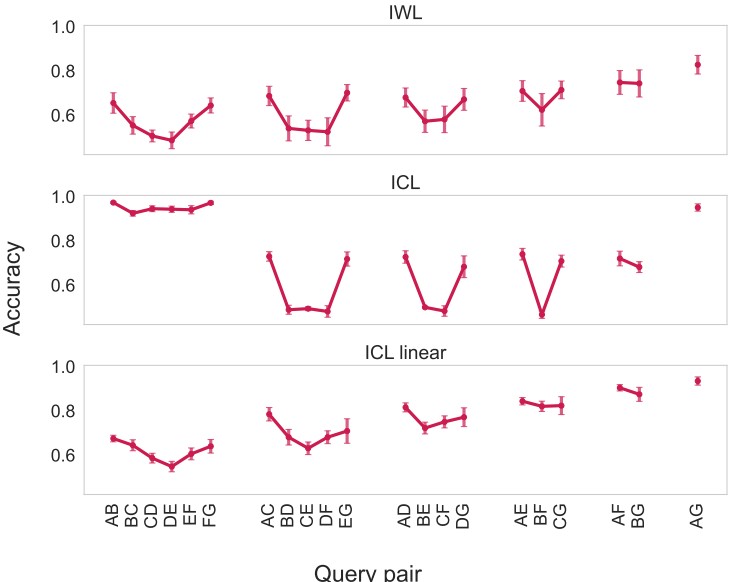

Figure A.1: Reproducibility across multiple runs. Performance (accuracy) on transitive inference task across 10 independent runs with different random seeds for in-weights learning (IWL, upper panel), standard in-context learning (ICL, middle panel), and linear regression pre-trained ICL (ICL linear, lower panel) models. Points show mean accuracy for each query pair, with error bars representing 95% confidence intervals. Results demonstrate consistent emergence of transitive inference and symbolic distance effects in IWL and linear pre-trained models, while standard ICL models consistently fail to generalize beyond adjacent pairs and terminal items.

## A.9 LARGE LANGUAGE MODEL EXPERIMENT SETUP

### A.9.1 DATASET AND PROMPT GENERATION

We evaluated large language models using the RECOGLAB transitive inference dataset (Liu et al., 2025), which contains 1000 examples each of four conditions:

1. **Congruent**: Items ordered consistently with real-world knowledge (e.g., "Sea turtle is smaller than Stage")

2. **Incongruent**: Items ordered opposite to real-world knowledge (e.g., "Pill is larger than Tomato")

3. **Random strings**: Meaningless letter-number combinations with no semantic content (e.g., "UpnuIS is smaller than 3i6QPs")

4. **Permuted**: Real items with shuffled relationships

Each example presents a hierarchy of 20 items through pairwise comparisons, followed by a query about two items' relative ordering.

### A.9.2 GEOMETRIC PROMPTING CONDITIONS

We augmented each base question with three prompting strategies:

- **Baseline**: Direct question with instruction "Answer yes or no"
- **Number line** (linear geometry): "Imagine all items lie on a number line from smallest to largest"
- **Circle** (non-linear geometry): "Imagine all items lie on a circle from smallest to largest"

### A.9.3 EXPERIMENTAL CONSTRAINTS

To isolate the effect of geometric scaffolding from explicit reasoning, we:

- Appended "Answer yes or no" to all prompts
- Limited model outputs to prevent chain-of-thought reasoning in non-CoT conditions
- Explicitly instructed models to provide only yes/no answers without explanation

This resulted in 12,000 total test instances (4 conditions $\times$ 3 prompting strategies $\times$ 1000 examples). Models were evaluated on accuracy of their yes/no predictions, with analyses focusing on how different geometric prompts affected performance across item conditions.

### A.9.4 MODEL EVALUATION

We evaluated all generated questions using Google Gemini's API on the following model labels: *gemini-2.5-pro*, *gemini-2.5-flash*, and *gemini-2.5-flash-lite*. We also evaluated local copies of *gemma3-4b-it*, *gemma3-12b-it*, and *gemma3-27b-it*. In total, 32,000 questions were answered for each model. For *gemini-2.5-pro* and *gemini-2.5-flash*, we disabled native thinking by setting token thinking budgets to 0. For OpenAI's *gpt-5*, *gpt-5-mini*, and *gpt-5-nano*, we set both reasoning and verbosity to their lowest possible setting.

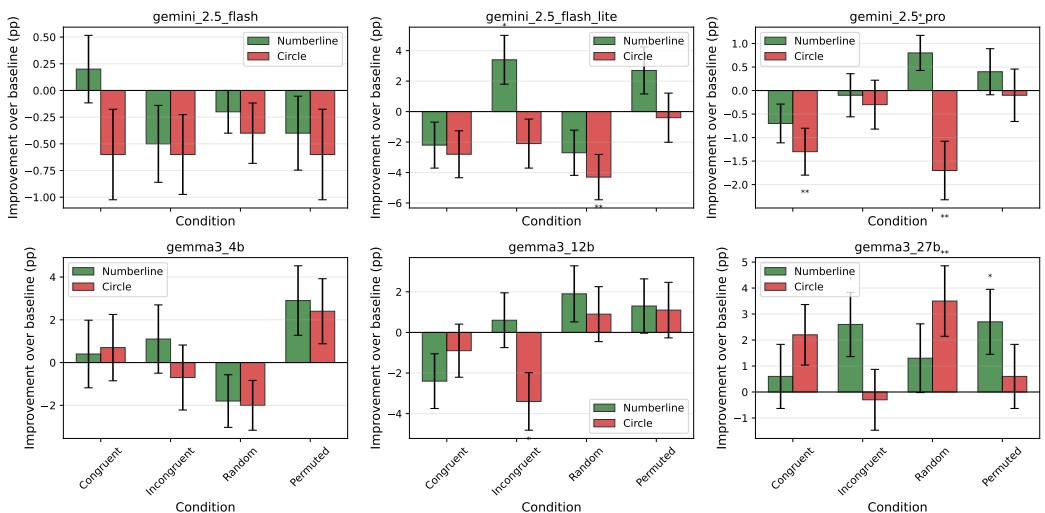

Figure A.2: Prompt effectiveness by item type (congruent, incongruent, random, and permuted conditions) for each model type separately. All improvements are measured in percentage points relative to baseline performance.

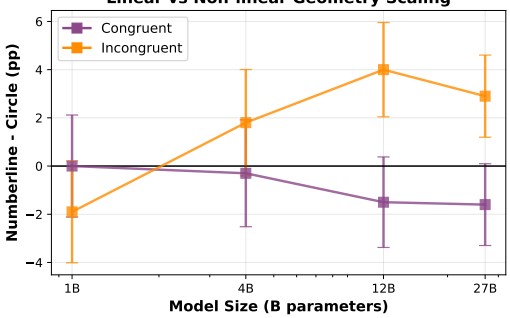

Figure A.3: Effect of model size on the advantage of number line over circle prompts for Gemma models. The performance difference between geometric prompts increases with model size for incongruent item pairs but remains constant for congruent pairs. Error bars represent SEM.

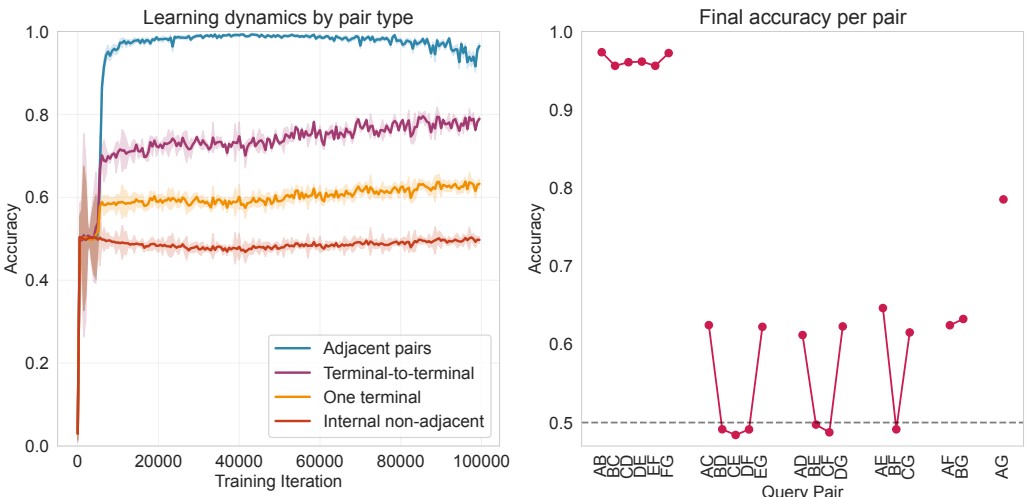

Figure A.4: Results for in-context learning experiments with the full transformer architecture with MLPs. In this setup, as in Figure 3, only adjacent pairs were chosen as queries during training, but we evaluate on all pairs. Left panel shows accuracy across iterations per pair type, while the right panel shows the final accuracy for each pair. Despite addition of MLP layers model did not show transitive generalization, as shown by the chance accuracy for non-adjacent pairs that did not include a terminal item.

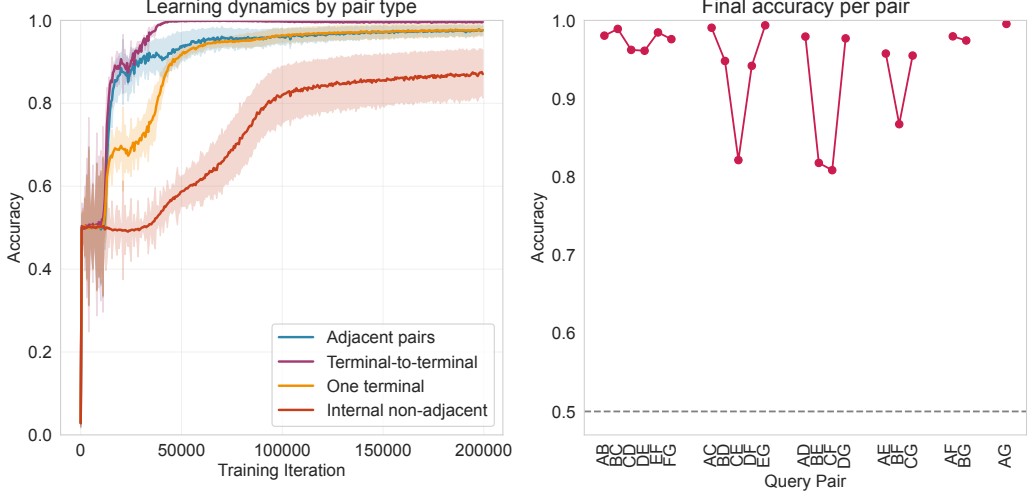

Figure A.5: Results for the in-context learning experiments with attention-only transformers, in a training condition where non-adjacent query pairs are added during training. While learning to generalize to non-adjacent pairs takes longer than learning adjacent and terminal items, the model learns transitive generalization (as shown by above chance performance for internal, non-adjacent pairs). Note that, in contrast to IWL and linear pre-trained models (Figures 3 and 5), this training regime does not result in a "symbolic distance effect".

