# OpenReview forum: "Examining relational reasoning and inductive bias in transformers trained on a transitive inference task"
_ICLR.cc/2026/Conference — Submitted to ICLR 2026_

### Official Review · Reviewer_bNc6 · 2025-10-31

**Soundness:** 2
**Presentation:** 3
**Contribution:** 2
**Rating:** 4
**Confidence:** 4

**Summary:**

This work explores how transformers learn a transitive inference task under in-weights learning versus in-context learning. For in-context learning, the input is a sequence of pairs of objects and their order relation, followed by a query for a pair of objects, and the target output is the order relation. For in-weights learning, the context sequence is replaced by random noise, and the model directly learns the order relation from the query pairs. For ICL, the ordering is randomized for each sequence to encourage a general in-context strategy rather than memorization. For the IWL setup, the ordering is fixed during training. In both cases, the queries in training data include only adjacent pairs (which, in the ICL setup, must appear in the context sequence).

The authors find that the IWL model generalizes to non-adjacent query pairs, whereas the ICL model does not. They show that the ICL model learns an induction head mechanism, explaining its inability to generalize. Motivated by previous work on linear models and transitive inductive biases, they pre-train the ICL model on an in-context linear regression task and find that this enables transitive generalization. Finally, this work seeks to connect these insights to large-scale models by exploring how prompting LLMs to "imagine" a linear vs circular structure affects performance on the ReCogLab task.

**Strengths:**

- Research problem & Motivation: The research question is well-motivated and grounded in cognitive science. The discussion of symbolic distance and terminal item effects is insightful, and observing these patterns in trained transformers is interesting.
- Integration of different analysis methods: The paper thoughtfully integrates empirical performance results with mechanistic interpretability analyses, which are both needed for understanding the inductive biases of different models under different training regimes.
- Model interpretability: the mechanistic analysis in the (no pretraining) ICL model identifies the learned computational circuit by testing the hypothesis that an induction head mechanism is being implemented, through attention head ablations and relating PCA embeddings to separability of the target. This explains the failure to generalize transitively.
- Clarity of presentation: the paper is well-written and provides interesting discussion, bridging literature from machine learning and cognitive science.

**Weaknesses:**

**Major Concerns:**

- One of the paper’s central claims is that the ICL model fails to generalize transitively, whereas the IWL model succeeds. However, this may partly reflect differences in the specific training setups rather than a fundamental limitation of in-context learning. In the ICL configuration, training queries always correspond to pairs that appear in the context sequence, making the simplest strategy to copy answers rather than infer broader relational structure. In other words, the problem could be that the model "doesn't know what the task is", rather than it failing to learn it (the training regime makes it appear like the task is to copy information from the context). While the authors’ intent is to test generalization to longer transitive chains, this could be achieved in other ways (e.g., train on queries with $|i - j| \leq k$ and evaluate on $|i - j| > k$, for $k > 1$). In short, the observed differences between IWL and ICL may stem from task design rather than from intrinsic properties of the learning paradigms themselves.
- The use of an attention-only transformer limits the applicability of the conclusions to real Transformers. The two types of models have very different types of inductive biases; this is an important limitation since the paper aims to make claims about the inductive biases of Transformers. For example, an MLP may be required for certain types of computation that would otherwise be learned by a Transformer (e.g., combining the results of multiple attention heads in the ICL configuration to infer the order relation for $|i -j| > 1$). While model interpretability is a valid concern, modern interpretability tools (e.g., sparse autoencoders) would enable interpretability of MLP components, especially in such simple synthetic settings.

**Other limitations:**
- The number of objects in the context, $N=7$, is quite small, making the scope of the experiments somewhat limited. It would be interesting and valuable to evaluate how the results scale with larger $N$.
- The mechanistic interpretability analysis for the linear regression-pretrained model is limited. It only shows that the learned model is not an induction head, but it does not identify the computational circuit that allows the model to generalize transitively. It would be interesting to know what this ICL model is doing that allows it to generalize.
- Generally, the task being studied is quite simple and synthetic, with a small scale ($N = 7$), yet the paper often makes broad claims relating to "relational reasoning" at large. For example, the claim that pre-training on linear regression can help with relational reasoning generally seems to be a big jump from the specific synthetic TI task considered in the paper.
- The connection between the linear regression pretraining and the LLM prompting experiments is unclear. The "linear geometry" prompt ("imagine a number line") seems conceptually distinct from the linear regression pretraining task, and the mechanisms driving performance improvements likely differ.

**Questions:**

- Have you tested setups where the ICL model is trained on queries not restricted to pairs appearing in the context? How would this affect generalization to $|i - j| > 1$?
- This work explores how the inductive biases of Transformers affect relational learning for IWL vs ICL. There has been prior work that explores architectures with explicit relational inductive biases, including some recent work on variants of Transformers (e.g., RelationNet, PrediNet, CoRelNet, Abstractor, etc.). I'd be interested in your thoughts about how these types of explicit relational inductive biases (which Transformers don't necessarily have) would fit into the picture.
- Can you elaborate on the mechanistic connection between linear regression pretraining and the TI task? *Why* does linear regression support transitive inference? Adding a brief intuitive summary (e.g., from the exploration in Lippl et al. (2024)) would strengthen this point in the paper.
- I was unsure how to read the figures with "Prediction Value" on the y-axis. I was expecting this to be an accuracy, like in Figure 1D on results from rhesus macaques, reprinted from Lippl et al. (2024). The paper says the IWL model performs "above chance". What are the accuracies achieved here in each configuration? It would be somewhat interesting to compare this to the data from rhesus macaques.
- One notable (potential) advantage of the ICL setup is the capacity for generalization to unseen objects. I.e., IWL will only work on the objects it sees during training (because it needs to learn their order/rank), whereas ICL can (in principle) generalize not only to new orderings but also to completely different objects/images. Can you comment on this? Is this something you explored?


Overall, this study offers an interesting perspective on a timely topic, and I look forward to further discussion with the authors.

---

> ### Author Response · Authors · 2025-11-20
>
> We thank reviewer bNc6 for their thoughtful comments. Firstly, we apologize for the mis-formatted text in figures 1, 2 and 3. We have now updated the pdf and we hope that this now shows the figures as they are.
>
> **Major Concerns**
>
> ICL training setup and task design: We want to emphasize that we are *not* claiming ICL has a fundamental limitation in performing TI. Rather, our findings demonstrate that ICL trained exclusively on adjacent pairs does not automatically develop transitive generalization, whereas IWL does, revealing a difference in inductive bias.
>
> The reviewer is correct that *"the training regime makes it appear like the task is to copy information from the context."* However, this is analogous to how TI experiments are designed in cognitive science: subjects are never asked to generalize to non-adjacent pairs during training, yet the key question is whether they spontaneously do so at test time. What we are investigating is the *inductive bias* of the system: what kind of generalization emerges naturally when trained only on adjacent pairs?
>
> Our linear regression pre-training experiments (Section 3.3) and LLM results (Section 4.3) demonstrate that ICL *can* achieve transitive inference when provided with appropriate structural scaffolding. To emphasise this point further, we have now also added experiments with non-adjacent queries during training (see below).
>
> *Attention-only architecture*: This is a fair point. We omitted MLPs primarily for interpretability, which is common practice in mechinterp studies. We have now conducted experiments with MLPs and find that the qualitative results remain unchanged: IWL models generalize transitively, standard ICL models do not (Figure A.4). While we cannot conduct a full sparse autoencoder analysis within the revision timeframe, we believe these additional experiments demonstrate that our core findings are not artifacts of the attention-only architecture.
>
> **Other Limitations**
>
> *Number of objects (N=7)*: We matched this to the cogsci literature on TI (e.g., Jensen et al., 2015; Lippl et al., 2024), which typically uses similar hierarchy sizes. We agree that studying chains with larger N is an interesting future direction.
>
> *Mechanistic analysis of linear pretrained model*: We agree that identifying the full computational circuit would be valuable. Previous work by von Oswald et al. (2023) has shown that transformers on this linear regression task implement gradient descent in the forward pass, so we expect similar mechanisms here.
>
> *Scope of claims about relational reasoning*: We appreciate this point and have reviewed our language. We are not claiming that linear regression pre-training would help relational reasoning in general. Rather, our interpretation is that exposure to the right structural scaffold (linear ordering) enables transitive inference, while incompatible structures (circular geometry) can impair it. We have clarified this in the revised manuscript.
>
> *Connection between linear pre-training and LLM prompting*: We acknowledge these may involve different mechanisms. The linear regression pre-training builds representations through gradient-based learning, while geometric prompting may activate latent capabilities developed during pre-training. However, both manipulations involve providing linear structural scaffolds, and both improve transitive inference performance. The toy model experiments inspired the LLM experiments, which revealed an interesting and consistent pattern.
>
> **Questions**
>
> *Training on non-adjacent pairs*: We have now conducted experiments where the model is trained with non-adjacent pairs included in the queries (Appendix Figure A.5). As expected, the model shows increased transitive generalization. This supports our interpretation that the difference between IWL and ICL reflects inductive biases rather than fundamental capabilities.
>
> *Architectures with explicit relational inductive biases*: This is an interesting direction. We would expect that architectures explicitly designed for relational reasoning (RelationNet, PrediNet, etc.) would perform well on in-context TI.
>
> *Mechanistic connection between linear regression and TI*: Thank you for this suggestion. We have added an intuitive explanation to the paper (lines 295-300).
>
> *Prediction values vs. accuracy*: Thank you for this feedback. In the original submission, following other studies of transitive inference, we displayed continuous prediction values. However, we agree with the reviewer that accuracy is more directly comparable to behavioral data, and more interpretable. In the revised manuscript, we have updated all figures to report accuracy instead of raw prediction values, like for the rhesus macaque data.
>
> *Generalization to unseen objects*: In our ICL evaluation, we do test on completely novel items: each sequence uses randomly sampled Omniglot images that were not seen during training. This is indeed a key advantage of the ICL setup, as you note.

---

> > ### Comment · Reviewer_bNc6 · 2025-11-26
> >
> > I thank the authors for their responses. I have reviewed the rebuttal and the updated manuscript.
> >
> > Below is an item-by-item response to the authors’ rebuttal.
> >
> > ---
> >
> > **Task design**
> >
> > I agree with the authors that IWL and ICL have different inductive biases. This is demonstrated in this paper, but is also a very intuitive and unsurprising claim. The key issue is the precise claim about *how* these inductive biases differ. My concern is that the conclusions are an artifact of somewhat arbitrary decisions in the task setup. In particular, as the authors acknowledge, the ICL training regime makes it appear like the task is to copy information from the context. This is further indicated by the results of the added experiments in Figure A.5, which show that transitive generalization *is possible* when the training regime does not only include adjacent pairs (which makes the task appear like copying information from the context).
> >
> > > The reviewer is correct that "the training regime makes it appear like the task is to copy information from the context." However, this is analogous to how TI experiments are designed in cognitive science: subjects are never asked to generalize to non-adjacent pairs during training, yet the key question is whether they spontaneously do so at test time. What we are investigating is the inductive bias of the system: what kind of generalization emerges naturally when trained only on adjacent pairs?
> >
> > I don't find the analogy to cognitive science experiments particularly convincing. It is clear that humans have different inductive biases to artifical models; for example, subjects will have some assumptions about what the task is because they know the task is designed by a human. A good experimental design for understanding the inductive biases of artificial models (e.g., IWL vs ICL) can be different from a good experimental design for cognitive science experiments.
> >
> > ---
> >
> > *Attention only architecture*
> >
> > > This is a fair point. We omitted MLPs primarily for interpretability, which is common practice in mechinterp studies. We have now conducted experiments with MLPs and find that the qualitative results remain unchanged: IWL models generalize transitively, standard ICL models do not (Figure A.4). While we cannot conduct a full sparse autoencoder analysis within the revision timeframe, we believe these additional experiments demonstrate that our core findings are not artifacts of the attention-only architecture.
> >
> > Thank you for adding theese experiments. This partially addresses this limitation. I would suggest performing a more thorough interpretability analysis for future versions of this paper if interpretability is positioned as a contribution of this work.
> >
> > ---
> >
> > *On the limited scale of the experiments*
> >
> > > Number of objects (N=7): We matched this to the cogsci literature on TI (e.g., Jensen et al., 2015; Lippl et al., 2024), which typically uses similar hierarchy sizes. We agree that studying chains with larger N is an interesting future direction.
> >
> > Thank you. I think following the cognitive science literature too closely can limit exploration of artificial models. Robustly studying the differences in inductive biases between IWL and ICL in Transformers will likely require different methods. One key advantage in artificial settings is that scaling up is feasible. Understanding how the conclusions behave under scaling is important for building a more complete picture.
> >
> > ---
> >
> > *Scope of claims about relational reasoning:*
> > > We appreciate this point and have reviewed our language. We are not claiming that linear regression pre-training would help relational reasoning in general...
> >
> > Thank you. I think this is important to ensure accuracy and soundness in the paper's claims.
> >
> > ---
> >
> > *Connection between linear pre-training and LLM prompting:*
> > > We acknowledge these may involve different mechanisms. The linear regression pre-training builds representations through gradient-based learning, while geometric prompting may activate latent capabilities developed during pre-training....
> >
> > I suggest making the difference in mechanisms explicit in the paper.
> >
> > I personally found the LLM evaluation section with these different prompts to be distracting and not very interesting.It felt out of place because the mechanism seems unrelated, and in my view the LLM prompt evaluation is not intrinsically compelling here. It can be tempting to include a section like this to approach the types of "large-scale experiments" that reviewers often ask for in machine learning venues, but I would suggest considering removing this section and replacing it with more detail and discussion on the core paper (perhaps with some of the additions discussed above).
> >
> > --
> >
> > Thanks again to the authors for their efforts, and I hope some of this feedback might be helpful to them.

---

> > > ### Author Response · Authors · 2025-12-02
> > >
> > > We thank Reviewer bNc6 for their continued engagement with our work.
> > >
> > > **On using cognitive science paradigms for understanding artificial models**
> > >
> > > We respectfully disagree that cognitive science-inspired experimental designs are inappropriate for understanding artificial systems. This methodology has been successfully employed across numerous influential works studying transformers and LLMs, including work by Lampinen et al. (2023), Binz & Schulz (2023), Webb et al. (2023), and others who have used cognitive tasks precisely because they are designed to reveal systematic patterns in reasoning and generalization. The value of these paradigms lies not in assuming artificial models share human inductive biases, but in their careful construction to isolate specific reasoning capabilities.
> > >
> > > Transitive inference tasks are designed to measure and characterize generalization behavior along systematically controlled variables. This is equally valuable in ML contexts as a starting point for understanding how learning and generalization in relational reasoning emerge in transformer models. While we agree that not all cognitive science paradigms translate meaningfully to ML settings, TI represents a case where the experimental logic (testing whether a system spontaneously generalizes beyond its training distribution) applies directly to understanding model capabilities.
> > >
> > > The reviewer suggests our results may be intuitive, but to our knowledge, the differential inductive biases between IWL and ICL for relational reasoning have not been systematically documented before. Characterizing this profile in transformers, and identifying the mechanistic basis (induction heads vs. geometric representations), constitutes a concrete contribution even if the high-level finding aligns with the reviewer's intuition.
> > >
> > > **On LLM experiments**
> > >
> > > We note that Reviewer wcp7 explicitly highlighted the inclusion of LLM experiments as a strength, stating that “the paper contains both experiments on transformers trained from scratch and extant large language models, which helps support a broader range of conclusions than either of those experiments in isolation." Additionally, Reviewer 1g7M's criticism that the paper has "limited importance for real LLMs" (apparently missing Figure 6, which we highlighted in our response) indicates that LLM experiments are valued by other reviewers as strengthening the paper's relevance. Given this feedback, we believe retaining these experiments serves the paper's goal of building understanding across model scales. We will, however, make the difference in mechanisms between pre-training and prompting more explicit as suggested.

---

### Official Review · Reviewer_wcp7 · 2025-10-31

**Soundness:** 3
**Presentation:** 2
**Contribution:** 2
**Rating:** 4
**Confidence:** 3

**Summary:**

The authors present an analysis of the ability for transformer models and large language models to perform transitive inference (i.e. inferring from examples of the general form “A > B” and “B > C” that “A > C”). Specifically, the authors compare a transformer model trained from scratch on fixed transitive inference data (allowing the model to store item relationships in weights) to a model trained on randomly-generated transitive inference sequences provided in-context (preventing memorization) and find that the first model successfully generalizes to non-adjacent query items while the second model does not. The authors also find evidence for an “induction circuit” in the second model which may explain its memorization-based behavior. Finally, the authors demonstrate that prompting LLMs with “linear” geometry improves their performance on transitive inference tasks compared to prompts with “circular” geometry (which does not support transitivity).

**Strengths:**

For the most part, this paper is clearly written and argued. The authors present a thorough set of experiments for teasing apart the differences between in-weight and in-context training with regards to performance on transitive inference. I appreciate that the paper contains both experiments on transformers trained from scratch and extant large language models, which helps support a broader range of conclusions than either of those experiments in isolation. The authors have also seemed to engage with much of the existing literature on the topic.

**Weaknesses:**

As I’m sure the authors are aware, Figures 1, 2, and 3 are seriously malformed and more-or-less impossible to read. This should obviously be corrected in any future versions of the manuscript, but I don’t count it as a particularly serious critique since it seems very easily fixable.

More seriously, I think the paper would benefit from a stronger argument as to the impact and implications of the work. The authors have successfully demonstrated that differences exist between different training types in terms of their transitive inference performance, but I would like to see a bit more discussion of what this might say about the capabilities of modern models, especially since they presumably exhibit effects of both in-weight and in-context learning. For instance, Figure 6B seems to indicate that circular geometry prompting improves the performance of the largest model despite the fact that the context doesn’t support transitivity. What might account for this? More generally, how might these results scale to more complicated problems of interest that have transitivity as a component? I'm open to arguments about the significance of these results and interested to hear the authors' perspective.

**Questions:**

- In Figure 2A, the characters in the “context” section appear to not match the english characters in the blue box above. Is this deliberate?
- In Figure 2D, the prediction values seem to range between 1 and 3. My understanding was that the model was trained to predict only the direction of the relationship (i.e. {-1, 1}) -- am I misunderstanding?
- Line 195: the text indicates that the terminal items are A and E, but the figure seems to indicate that A and G are terminal items, while E is non-terminal -- which is correct?
- Just to clarify: during in-context training, the model is still only presented with adjacent queries, correct? So the “accuracy” Figure 3C is referring only to adjacent-pair accuracy?
- Line 454: The reference to Figure 9 should probably be to Figure 6 instead
- A potentially relevant citation: “Rane, Sunayana, et al. "Position: Principles of Animal Cognition to Improve LLM Evaluations." Forty-second International Conference on Machine Learning Position Paper Track.”

---

> ### Author Response · Authors · 2025-11-20
>
> We thank reviewer wcp7 for their thoughtful comments. Firstly, we sincerely apologize for the missing and mis-formatted text in figures 1, 2 and 3. This seems to be a compilation error, which did not occur during the test compilation at submission, so we suspect there was a technical error. We have now updated the pdf and we hope that this now shows the figures as they are.
>
> **Impact and Implications**
>
> We appreciate the reviewer's suggestion to strengthen our discussion of impact and implications for modern models. We have substantially expanded the Discussion section to address these points.
>
> Regarding the intriguing observation about circular geometry prompting in Figure 6B: We believe that for some models, geometric prompts in general (even circular ones) may help by activating a reasoning mode focused on relationships rather than relying solely on stored knowledge. We observe this effect most notably in some of the larger open-weight models like Gemma 27B, though we note that the proprietary models (Gemini series) are likely substantially larger than Gemma 27B.
>
> However, the critical finding is that across all models except Gemma 1B (which is likely too small to effectively interpret geometric prompts), the number line prompt consistently outperforms the circular prompt. This advantage is most pronounced in the incongruent and permuted conditions (Figure 6C), where models cannot rely on prior knowledge and must perform genuine in-context reasoning. In these conditions, the circular geometry (which violates transitivity) maximally interferes with the task. In contrast, for congruent items where stored knowledge aligns with the task, both geometric prompts show smaller effects because the model can partially rely on information in its weights. This pattern parallels our trained-from-scratch findings: when models must use in-context learning rather than stored knowledge, the structural compatibility of the geometric scaffold becomes critical.
>
> More broadly, our results have implications for understanding modern LLMs, which exhibit both in-weights learning (from pre-training) and in-context learning capabilities. Our findings suggest that: (1) the training regime fundamentally shapes which inductive biases emerge, (2) appropriate structural scaffolding (whether through pre-training on tasks with linear structure or through geometric prompting) is crucial for transitive inference, and (3) models may default to simpler pattern-matching strategies (like induction circuits) when the training distribution allows it, even when more general relational inference would enable better generalization. For complex real-world problems involving transitivity as a component (e.g., temporal reasoning, social hierarchies, preference learning), our results suggest that prompting strategies that activate appropriate geometric or relational structures could substantially improve performance, particularly when the task requires reasoning beyond stored knowledge.
>
> We have expanded the Discussion section to elaborate on these points and their implications for understanding modern language models.
>
> **Questions**
>
> *Figure 2A context characters*: Correct. In the IWL experiment, we matched the context length to the ICL condition for consistency, but the context contains random unrelated image pairs that provide no information about the hierarchy. This forces the model to store the rank order in its weights. We have clarified this in the revised text.
>
> *Figure 2D prediction values*: Thank you for catching this. Following previous work on computational models of TI (e.g., Lippl et al., 2024), we trained the model using MSE loss on continuous predictions, which are then thresholded at zero for classification (positive predictions indicate item 1 > item 2, negative predictions indicate item 1 < item 2). The figure showed these continuous prediction values before thresholding. However, since this caused some confusion among multiple readers, we have now switched to reporting accuracy throughout.
>
> *Line 195 terminal items*: Yes, A and G are the terminal items. We have corrected this error in the revised manuscript.
> Figure 3C accuracy: Yes. During in-context training, only adjacent queries are presented, so the training accuracy in Figure 3C refers to adjacent-pair accuracy. We have clarified this in the figure caption.
>
> *Line 454 figure reference*: The correct reference was to Figure 9 in the appendix, but on re-reading the text on line 454 we understand the confusion. For clarity we have made this clearer in the text on that line, which now reads: The fact that, within the open-weight models of known size, larger models show a bigger effect, supports this interpretation (Figure AX). We have also changed the figure numbering in the appendix to start from 0 with an A as prefix.
>
> *Citation suggestion*: We have added the Rane et al. citation, which is indeed highly relevant to our work on cognitive science principles and LLM evaluation.

---

### Official Review · Reviewer_EgBT · 2025-11-01

**Soundness:** 3
**Presentation:** 3
**Contribution:** 2
**Rating:** 4
**Confidence:** 3

**Summary:**

The paper studies whether transformers can learn simple relational rules in context using a transitive inference task (the ability to infer that if A > B and B > C, then A > C). The authors train small two-layer attention-only transformers on a task where the model sees short sequences of pairwise comparisons between symbols. Each item is an Omniglot images passed through a frozen ResNet18. The model’s input consists of triples: two item embeddings and a label indicating which one ranks higher. Importantly, during training the models only ever see adjacent pairs. At test time, it must predict relations for non-adjacent pairs.

The authors compare two training regimes: (1) in-weights learning (IWL), where the hierarchy of items is fixed across all sequences such that the model can memorise the order in its parameters, and (2) in-context learning (ICL), where a new random hierarchy is drawn for each sequence, forcing the model to infer the order from the examples in the context window. They find that models trained with IWL generalise to non-adjacent pairs whereas ICL models fail completely on non-adjacent pairs. An analysis of attention patterns suggests that induction heads play a crucial role in the ICL model. In other words, the model just learned to look up solutions in the context.

To test whether this limitations is a result of missing representational structure, the authors pre-train the same model on an in-context linear-regression task, which plausibly builds a one-dimensional latent geometry, and then fine-tune it on the transitive inference task. After this pre-training, the model is capable of generalising to non-adjacent pairs and does not rely on induction heads anymore. Finally, large language models show a similar pattern: they solve transitive inference-style prompts more reliably when the relations are described along a line than when arranged on a circle.

**Strengths:**

- The paper addresses an interesting question about the generalization limits of in-context learning.
- The experiment showing a benefit of pre-training on linear regression is interesting and novel.

**Weaknesses:**

- All mechanistic conclusions (e.g., induction circuits versus distributed geometry) are drawn from a two-layer, attention-only transformer without MLPs. Since MLPs play a central role in representation building in standard transformers, removing them may substantially alter model behavior. The observed ICL limitations might therefore be specific to this simplified architecture.
- The conclusion that linearity in the regression pretraining task is the key enabling factor is not strongly supported. Other properties, such as the need to integrate information globally across the context, might equally explain the improvement.
- The use of fixed ResNet18 embeddings introduces unnecessary high-dimensional structure that may not be relevant to the symbolic nature of the task. Testing with simpler, low-dimensional or one-hot embeddings would clarify whether the results depend on this choice.

**Questions:**

- How robust is the ICL limitation? If the training data included a small proportion of non-adjacent pairs (e.g., 5%), would the model begin to form a distance-like or global representation, or would it continue relying on match-and-copy strategies?
- Could you clarify whether MLPs were omitted primarily for interpretability or for computational reasons, and whether adding them changes the qualitative results?

---

> ### Author Response · Authors · 2025-11-20
>
> We thank reviewer EgBT for their thoughtful comments. Firstly, we sincerely apologize for the missing and mis-formatted text in figures 1, 2 and 3. This seems to be a compilation error, which did not occur during the test compilation at submission, so we suspect there was a technical error. We have now updated the pdf and we hope that this now shows the figures as they are.
>
> **Clarification of main claims**
>
> We want to emphasize an important distinction in our findings: we are not claiming that in-context learning is fundamentally incapable of transitive inference. Rather, our results demonstrate that ICL trained exclusively on adjacent pairs does not automatically develop transitive generalization, whereas IWL does. This difference in *inductive bias* is the core finding. Indeed, our linear regression pre-training experiments (Section 3.3) explicitly demonstrate that ICL *can* achieve transitive inference when provided with appropriate structural scaffolding. In addition, we have added a new figure (Appendix figure A.5) which shows that providing the model with non-adjacent queries during training results in successful transitive inference for ICL models. In this condition, we do not see a symbolic distance effect. We have clarified this distinction in the revised manuscript.
>
> **Weaknesses**
>
> *Attention-only architecture*: We acknowledge this is an important limitation. MLPs were omitted primarily for interpretability purposes, which is a common approach in mechanistic interpretability studies with small transformers (e.g., Olsson et al., 2022). However, we have now conducted additional experiments with MLPs and find that the qualitative results remain unchanged: IWL models generalize transitively, standard ICL models do not, and linear regression pre-training enables transitive inference in ICL. We have added these results to the revised manuscript, in Appendix Figure A.4.
>
> *Linear vs. global integration*: This is an important point. We agree that disentangling whether linearity specifically versus global integration more generally is the key factor would strengthen our claims. While our LLM experiments (Section 4.3) provide some evidence for the importance of linear structure specifically (where circular geometric prompts that violate transitivity impair performance) we acknowledge that additional controlled experiments (such as circular regression pre-training) would provide more direct evidence. We have added this as a limitation in the Discussion and as a direction for future work.
>
> *ResNet18 embeddings*: Our choice of ResNet18 embeddings of Omniglot images was motivated by their widespread use in few-shot and in-context learning studies (Chan et al., 2022; Reddy, 2024), making our setup comparable to prior work, and providing interpretable examples for the figures. However, we do not expect the specific embedding choice to affect our core findings, as the task fundamentally requires only that items be distinguishable. The transitive inference phenomenon we study depends on the relational structure between items rather than their specific representations. Future work could verify this with simpler embeddings (e.g., one-hot encodings), though we would expect qualitatively similar results.
>
> **Questions**
>
> *Robustness of ICL limitation*: We have conducted the suggested experiment where training data includes non-adjacent pairs, and show the results in Appendix Figure A.5. We find that with this exposure, ICL models do show transitive generalization on novel item sequences (but do not show a symbolic distance effect). This supports the interpretation that the ICL limitation is not fundamental but rather reflects the specific training setup. Importantly, this result reinforces our main point: the *inductive bias* differs between learning paradigms when trained only on adjacent pairs, with IWL naturally generalizing transitively while standard ICL requires additional structural information (either through non-adjacent training examples or appropriate pre-training). We have added these results to the revised manuscript.
>
> *MLP layers*: As noted above, MLPs were omitted for interpretability, following common practice in mechanistic interpretability research. We have now verified that adding MLPs does not change the qualitative results, which we report in the revised manuscript (Figure A.4).
>
> We believe these additional experiments substantially strengthen the paper and address the reviewer's concerns. We thank the reviewer again for their constructive suggestions.

---

> > ### Comment · Reviewer_EgBT · 2025-11-27
> >
> > Thank you for your response and additional experiments. My main concerns have been addressed and I will raise my score to 6 based on that.

---

### Official Review · Reviewer_1g7M · 2025-11-04

**Soundness:** 3
**Presentation:** 2
**Contribution:** 2
**Rating:** 4
**Confidence:** 3

**Summary:**

The paper studies in-context learning as well as trained transformers for a relational learning task.

**Strengths:**

Writing:
- Overall, the writing is clear and easy enough to follow, but with a number of caveats (see below).

Conceptual:
- The idea of looking for transitivity from ICL and a learned perspective is interesting.
- The result that ICL does not perform well whereas the learned model does is interesting as well.
- The found circuits shed a light on what is going on to some extent.

**Weaknesses:**

Writing:
- Figures 1 and 2 are made poorly. In each caption subfigures A), B), ... are referenced, but it is not clear to which of the images they refer to. Also, they are mostly some symbols that are hard to interpret. A prominent example is Figure 1: It shows in the upper left corner some letters, then diagrams that are completely non intelligible without looking at the caption, then some token sequences that are also cryptic. Illustrations are supposed to be as self-explanatory as possible.
- Also, I do not understand how the upper left image in Figure 1 is encoding a hierarchy. Also other parts of the image are poorly done and hard to understand.
- page 2: Acronym TI used before it is defined a few sentences below.
- Section 2.2 on the training setup is too vague and I would not be able to reproduce it from reading the paper. I would suggest to make it exact, precise and detailed enough to let the reader have a better understanding.
- Overall, the illustrations are done poorly and since they encode the results of the studies conducted in the paper, I do not think that the paper is fit for publication purely on presentational grounds.

Conceptual:
- The training setup seems somewhat convoluted and made-up. I am not sure this bears great relationship to reasoning tasks found in the real world.
- I am not sure that the setup makes sense as regards the in-context learning. ICL only emerges at larger scale in LLMs and hence I am unsure whether your setup with a smaller transformer trained from scratch adequately captures the properties of ICL as found in LLMs of interest.
- The analysis from the mechanistic interpretability point of view is not very deep, even though it contains some interesting patters in Section 3.2. Some circuits are identified, but no complete mechanistic understanding is achieved. The interpretation is just a standard application of a few tools without any deeper dive into the mechanics.
- In Section 3.3. I do not see how the claims are substantiated. The results in Figure 5 are not giving too much of an indication on that.
- Overall, the paper is a synthetic study with limited importance for real LLMs and limited mechanistic results for the synthetic setup they had, even though I think that one could obtain a more complete understanding for the simple 2-layer transformer for the studied task.

Other:
- No code and datasets were accompanying the paper and they are not announced to be released upon acceptance.

**Questions:**

- What is the connection between ICL in current LLMs and in your training setup. Can any insight be drawn from your study on that?

---

> ### Author Response · Authors · 2025-11-20
>
> We thank reviewer 1g7M for their thoughtful comments. Firstly, we sincerely apologize for the missing and mis-formatted text in figures 1, 2 and 3. This seems to be a compilation error, which did not occur during the test compilation at submission, so we suspect there was a technical error. We have now updated the pdf and we hope that this now shows the figures as they are.
>
> **Writing and presentation**
>
> We appreciate the feedback on clarity. We have corrected the early use of the TI acronym before its definition on page 2. Regarding Section 2.2, we have substantially expanded the training setup description to include precise details about hyperparameters, data generation procedures, and training protocols that will enable reproducibility.
>
> **Conceptual issues**
>
> *Ecological validity of the task*: We respectfully disagree that the training setup lacks relationship to real-world reasoning. Transitive inference is a fundamental cognitive task that has been studied extensively in cognitive science for decades across humans, primates, and other animals. While of course it does not capture all of relational reasoning, it captures a core component of it that underlies many real-world inferences. In our opinion this is interesting to study in isolation. Furthermore, in Section 4.3 and Figure 6 we relate our findings to modern large language models.
>
> *ICL in small vs. large models*: The reviewer states that “ICL only emerges at larger scales in LLMs”. This was a common takeaway from the GPT-3 paper in 2020, where the authors only saw ICL with larger models. But there they were measuring a specific type of few-shot learning that was dependent on first learning all the syntactic and semantic content from the natural language data. After this, in-context learning has been demonstrated in small transformers trained from scratch in a multitude of studies (Olsson et al., 2022; Chan et al., 2022; von Oswald et al., 2023; Akyürek et al., 2023; Singh et al., 2024). Furthermore, these studies have made a strong case that the mechanisms of ICL generalize across small models and LLMs (see Argument 6 in Olsson et al., 2022).
>
> Our results align with this literature, showing that ICL capabilities emerge even in models with relatively few parameters when trained appropriately. The key difference between IWL and ICL training regimes lies in whether the model can memorize a fixed hierarchy versus must infer it in-context, a distinction that holds regardless of scale.
>
> *Mechanistic interpretability depth*: We believe identifying the emergence of induction heads in the ICL condition versus their absence in IWL is a significant mechanistic finding that explains the qualitative difference in generalization behavior. However, we appreciate the reviewer's suggestion and would be happy to discuss what additional mechanistic analyses they would find valuable. Could the reviewer clarify what specific aspects of the mechanism they would like to see explored further?
>
> *Claims in Section 3.3*: Could the reviewer please specify which claims in Section 3.3 they feel are insufficiently substantiated by Figure 5? Our main claim in that section states “These results suggest that pre-training on tasks with inherent linearity promotes the development of representations that capture transitive relationships, rather than the pattern matching strategies observed in our earlier experiments.” Given the linear pre-training resulting in transitive inference, we feel that this claim is sufficiently substantiated by Figure 5, which shows that a linear regression pre-training paradigm has resulted in transitive inference behaviour as well as L2 representations that capture the linear structure.
>
> *Relevance to real LLMs*: As noted above, our work *does* include direct experiments with large language models (Section 3.4, Figure 6), which demonstrate how geometric structure in prompting affects their transitive inference performance. These LLM experiments were directly inspired by our mechanistic understanding from the small model studies, showing the practical relevance of our findings. In the discussion (section 4.3) we discuss bridging trained-from-scratch transformers and pre-trained LLMs. In addition, we have now added trained-from-scratch experiments with the full transformer architecture, including MLPs (Appendix figure A.4).
>
> *Code Release*: We will release code after acceptance.

---

### Meta-Review · Area_Chair_A1eq · 2026-01-05

**Summary:**

A main reason for my rejection recommendation is the issue of an incorrect author list (as noted by the authors during the rebuttal process). After discussing with the Senior AC, it would be impossible to modify the author list at this time because of possible conflicts, and we are therefore rejecting the paper. I strongly recommend that the authors __quadruple check__ their author list (after hitting submit, before the deadline) for the next submission. With the inclusion of the new results, and the correct author list, I believe a future submission will be accepted for publication.

Other issues that were brought up by reviewers:

__Presentation__
* several reviewers mentioned issues with the figure rendering. These were fixed during the rebuttal, and post-fixing, the figures look nice.

__Impact__
* reviewers were generally concerned about whether the findings of the study would be important / relevant for modern LLMs, given that the experiments presented mostly stripped down transformer architectures considerably (e.g. by removing MLPs). During the rebuttal, the authors mostly addressed these concerns by running experiments and confirming results for modified transformers which included MLPs.

__Experimental setting__
* some concerns were raised about whether the experiments performed were appropriate for transformers / LLMs, given that they were inspired by studies in cognitive science, which have a specific set-up in their prompting and in the number of entities / relationships they generally consider. While I agree with the authors that cognitive science studies are a useful and valuable way of framing experiments with LLMs, I also agree with the reviewer that it would be straightforward and valuable to see how the results scale when larger number of entities are considered, beyond those that have been studied in cognitive science. This is an especially valid point since the paper is not aiming to provide a modeling account of cognitive phenomena, it is simply an inspiration here for a capability to investigate (and how that capability might arise) in LLMs.

**Reviewer Concerns:**

__Addressed concerns__
* experiments with MLPs
* revising language throughout the paper to soften claims
* robustness of results -- including an experiment with 5% non-adjacent pairs changes the ability of ICL models to perform transitive inference

__Unaddressed concerns__
* scaling of experiments beyond the cognitive science prior work

**Reviewer Scores:**

I believe that all reviewers except for bNc6 would have raised their score to a 6. The authors wrote a strong rebuttal that addressed most points.

---

### Decision · Program_Chairs · 2026-01-26

Reject